# Finite Element Modeling of Acoustic Nonlinearity Derived from Plastic Deformation of 35CrMoA Steel

Shumin Yu [1], Lei Hu [2,*], Xingbin Yang [1] and Xiangyu Ji [2]

[1]   Zhengzhou Guodian Mechanical Design Institute Co., Ltd., Zhengzhou 450046, China; yusm@chec.com.cn (S.Y.); yangxb@chec.com.cn (X.Y.)
[2]   School of Materials Science and Engineering, Anhui University of Technology, Ma'anshan 243032, China; xiangyujiahut@sina.com
*   Correspondence: hulei828@ahut.edu.cn

**Abstract:** Acoustic nonlinearity derived from microstructural evolution of metallic materials during plastic deformation has been found to be a promising nondestructive technique to identify early stage plastic damage in metallic structural components. In the current investigation, the propagation of longitudinal ultrasonic waves in plastically deformed 35CrMoA steel plates was simulated using finite element (FE) methods based on the theory of dislocation-induced acoustic nonlinearity to establish the relationship between acoustic nonlinearity parameters and plastic strain. Experiments were conducted to validate the numerical model. Both simulated and experimental results demonstrate a monotonic increase in the acoustic nonlinearity parameter with applied plastic strain. The simulated ultrasonic nonlinear parameters deviate from experimental measurements in a two-stage pattern. In the low-strain regime (plastic strain < 8.5%), FE predictions underestimate experimental values, possibly due to dislocation entanglement in high-density regions that restricts dislocation mobility and suppresses acoustic nonlinearity. The FE model overestimates the parameters when plastic strain exceeds about 8.5%. This reversal is related to the formation of dislocation cells and walls with enhanced acoustic nonlinearity.

**Keywords:** 35CrMoA steel; ultrasonic nonlinearity; second-harmonic generation; plastic damage; finite element simulation

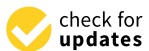

## 1. Introduction

Plastic deformation is a fundamental phenomenon in the manufacture and service of metallic materials. Plastic damage is generated with the accumulation of plastic deformation resulting from changes in microstructure, such as dislocation multiplication, annihilation, and subgrain initiation.

Failure occurs when the level of plastic damage exceeds a certain threshold [1–3]. Therefore, the quantitative detection of plastic damage is of great significance in predicting the residual service life of metallic structures. Nonlinear ultrasonics is considered to be one of the effective methods for assessing early-stage damage of materials, such as plastic damage, fatigue damage, and creep damage [4–6].

The ultrasonic nonlinearity arises from the nonlinear stress–strain relationship resulting from the evolution of microstructures, such as crystal lattices and dislocations. The magnitude of damage in metal materials can be characterized by cumulative plasticity resulting from changes in microstructure, such as dislocation density. The magnitude of ultrasonic nonlinearity exhibits distinct differences between intact and damaged material states. This discrepancy enables the evaluation of material degradation through ultrasonic

nonlinearity measurements, which are quantified by analyzing harmonic amplitudes in propagated ultrasonic waves [7–9]. To quantitatively describe the relationship between ultrasonic nonlinearity and material microstructure, a series of theoretical models have been established, including the dislocation string model, dislocation dipole model, and cumulative plastic strain model [10–13]. Hikata et al. [14] modeled the dislocation motion contribution to acoustic nonlinearity based on the dislocation string vibration model of Granato and Lücke [15] and experimentally verified in a high purity single crystal aluminum. The Hikata et al. model was later extended by Cash and Cai [16] to take into consideration orientation-dependent line energy and verified with dislocation dynamics simulations. Zhang et al. [17] improved the Hikata et al. model by taking into account the nonlinearity arising from screw and edge dislocations. However, it is still a challenge to capture the integrating quantitative microstructural contributions (e.g., dislocation density, precipitation, etc.) by experimental observation to establish the damage to constitutive relationships. Numerical simulations have been increasingly applied in ultrasonic nonlinearity studies due to their high efficiency and low cost. Some numerical models have been proposed to simulate the ultrasonic nonlinearity of media and/or damage, such as the finite element method (FEM) [18], finite difference time domain method (FDTD) [19], and local interaction simulation approach (LISA) [20]. Zhu et al. [21] simulated the longitudinal wave propagation in plastically deformed 30Cr2Ni4MoV martensite stainless steel and a monotonically increasing tendency of the acoustic nonlinearity parameter as a function of plastic strain was found. Wang et al. [22] modeled the effect of porosity on ultrasonic nonlinearity. The results showed that the relative nonlinear coefficient increases with the increasing pore length and decreases with the increasing pore width. Kim et al. [23] presents a non-destructive method for quantitative assessment of fatigue damage of materials with linear elastic properties. The semi-analytical finite element (SAFE) technique was used to obtain the phase-matching modes of the specimens. Li et al. [24] proposed a 2D hybrid spectral/finite element scheme for numerically resolving crack-induced contact acoustic nonlinearity in solid structures. Cao et al. [25] proposed a dedicated FE model to scrutinize the modulation mechanism of various modalities of pitting damage on the probing ultrasonic waves. Chen et al. [26] simulated the nonlinear mixed-frequency ultrasonic wave transmission mechanism from the cracks with different depths and the excited frequency. However, it is still a challenge to directly model the microscopic microstructure (e.g., dislocations, precipitates, etc.) of a macroscopic engineering structure due to the huge computational cost.

In the current investigation, a finite element model based on the dislocation-induced ultrasonic nonlinearity theory was established to simulate the ultrasonic nonlinearity of plastically deformed material. The Kocks–Mecking model was used to simulate the evolution of dislocation density. The ultrasonic nonlinear coefficients of material at different plastic damages were simulated to obtain the relationship between plastic damage, dislocation evolution and ultrasonic nonlinearity of the material. Subsequently, experimental validation was carried out to verify the accuracy of this model. The simulated results provide a theoretical foundation for the ultrasonic nonlinear evaluation of plastic damage of metal material.

## 2. Experimental Procedure

The material used in this study is 35CrMoA steel with the following chemical composition (wt. %): C 0.357, Cr 1.025, Mo 0.188, Si 0.212, Mn 0.688, P 0.015, S 0.0048, and Fe as the balance. The dimensions of the specimens are shown in Figure 1. Initially, the specimens were stretched to a predefined strain at room temperature using a universal testing machine with a controlled strain rate of 2.0 mm/min. A total of six groups of specimens

were prepared, each containing two specimens in each group. Group A0 represented the original specimen used as a reference while group A5 was stretched until it fractured to determine the maximum elongation. Groups A1 to A4 were stretched to 20%, 40%, ..., 80% of the maximum elongation, respectively, resulting in the corresponding strain values of 0.4%, 3.5%, 6.8% and 11.2%, respectively.

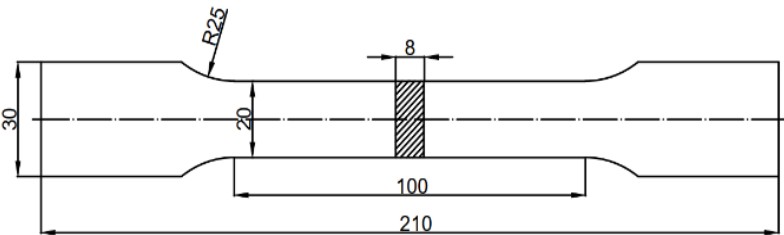

**Figure 1.** Dimensions of the specimen (Unit: mm).

Subsequently, the acoustic nonlinearity coefficients of both the original and plastically deformed specimens were measured using the RITEC SNAP RAM-5000 nonlinear ultrasonic testing system (RITEC Inc., Warwick, RI, USA). The experimental setup for nonlinear ultrasonic measurements is shown in Figure 2. A calibration step was conducted using a pulse-echo setup before the nonlinearity measurements to obtain the calibration function. The calibration function converts the electrical output of the receiving transducer into displacement amplitude to compensate for the coupling effect of the receiving transducer [27]. It is important to maintain the same contact condition between the specimen and the transducers in all tests. For this purpose, a specially designed fixture was used to secure the transducers, ensuring that a constant pressure of 3 kg × 9.8 = 29.40 N was applied in all tests.

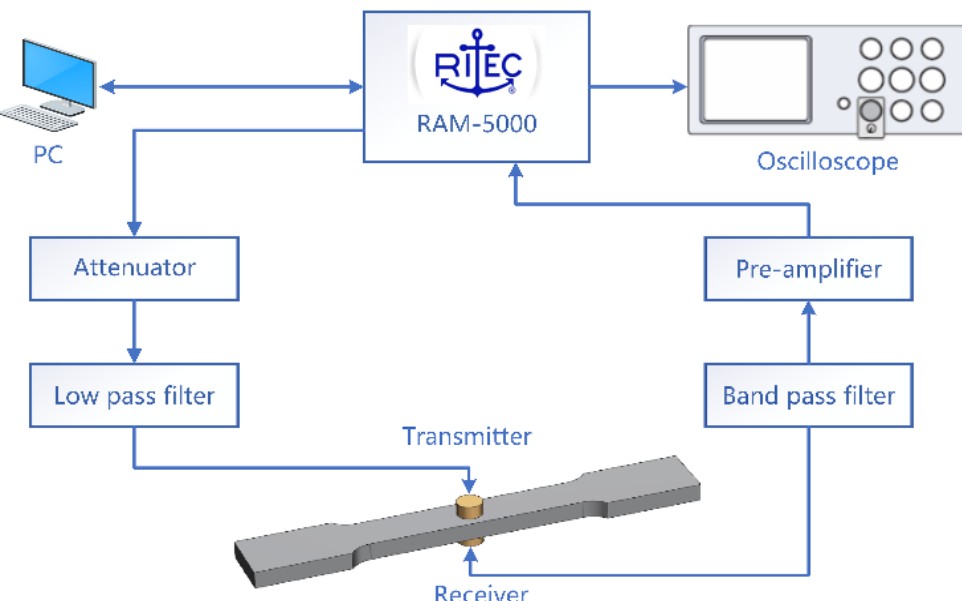

**Figure 2.** Experimental setup for nonlinear ultrasonic measurements.

To generate and measure the longitudinal wave, two commercial piezoelectric transducers with a diameter of $\Phi = 8$ mm were positioned at the center of the top and bottom surfaces of the specimens. A narrow-band longitudinal piezoelectric transducer was used to transmit a high-power fundamental wave while the receive transducer operated in pulse-echo mode and was excited with a broadband signal. The central frequencies of the transmitter and receiver are 5 MHz and 10 MHz, receptively, to the second harmonic

components of longitudinal wave. The cutoff frequencies of the low- and high-pass filters are 1 and 20 MHz, respectively. The transmitter is operated with a voltage of 450 V. A high-voltage tone-burst signal consisting of 5 cycles, with a spatial length of approximately 8.0 mm, is modulated using a Hanning window and then fed into the transmitter. The transmitted signal waveform is shown in Figure 3.

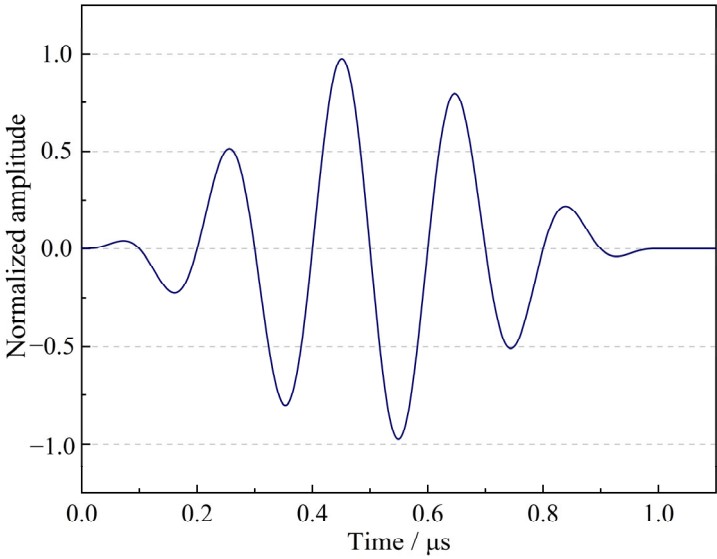

**Figure 3.** Waveform of the transmitted signal.

The one-dimensional nonlinear constitutive equation in an isotropic solid medium can be described as [28]:

$$\sigma_{xx} = E\varepsilon_x(1 + \beta\varepsilon_x + \ldots) \tag{1}$$

where, $E$ is the elastic modulus, $\sigma_{xx}$ is stress, $\varepsilon_x$ is the strain, $x$ is the material coordinate, and $\beta$ is the second-order nonlinear coefficient. When the strain is small, the relationship between strain and displacement is given by:

$$\varepsilon_x = \frac{\partial u}{\partial x} \tag{2}$$

where, $u$ is displacement. Ignoring the effects of scattering and attenuation, the wave equation of one-dimensional longitudinal wave in the solid medium is:

$$\rho\frac{\partial^2 u}{\partial t^2} = \frac{\partial \sigma_{xx}}{\partial x} \tag{3}$$

where $\rho$ is the density of the medium, and $t$ is time. By substituting Equations (1) and (2) into Equation (3) and ignoring the high order terms, the one-dimensional longitudinal wave equation in nonlinear media can be derived as:

$$\rho\frac{\partial^2 u}{\partial t^2} = E\frac{\partial^2 u}{\partial x^2} + 2\beta E\frac{\partial u}{\partial x} \cdot \frac{\partial^2 u}{\partial x^2} \tag{4}$$

Consider a single-frequency sinusoidal incident wave, namely $u = A_1\sin(\omega t)$, where amplitude of fundamental wave is $A_1$ and $\omega$ is angular velocity. According to the perturbation theory, the solution of the wave equation is [29]:

$$u = A_1\sin(kx - \omega t) - \frac{1}{8}\beta k^2 A_1^2 x\cos 2(kx - \omega t) \tag{5}$$

where, $k$ is the wave number of the propagating wave. According to Equation (5), amplitude $A_2$ of the second harmonic is:

$$A_2 = \frac{1}{8}\beta k^2 A_1^2 x \tag{6}$$

Then, the second-order nonlinear coefficient of the material can be expressed as:

$$\beta = \frac{8}{k^2 x} \frac{A_2}{A_1^2} \tag{7}$$

According to Equation (7), the second order acoustics nonlinear coefficient of the material can be obtained through the Fast Fourier transform (FFT) transformation of the experimentally measured time domain received signal. Subsequently, the amplitudes $A_1$ and $A_2$ of the fundamental wave and the second harmonic in the received signal can be determined. To minimize the influence of ultrasonic near-field effects, the secondary reflected echo from both the lower and upper surfaces of the specimen was selected for measuring the second order acoustics nonlinear coefficient. The frequency scan range of the received signals begins at 1 MHz and extends up to 20 MHz. Each received signal file comprises approximately 1200 data points, providing detailed information at various frequencies. Additionally, a post-averaging process was applied to the harmonic signals to enhance stability. This process ensures that the measured values accurately reflect the overall level of the nonlinear effect present in each plastically deformed specimen. Through post-averaging, the values are smoothed and become more consistent, thus improving the reliability of the results. To reduce the testing error, each sample was measured three times. Figure 4 illustrates one of the frequency spectrum curves of the received signal of the Group A4 specimen, highlighting the features of received signal.

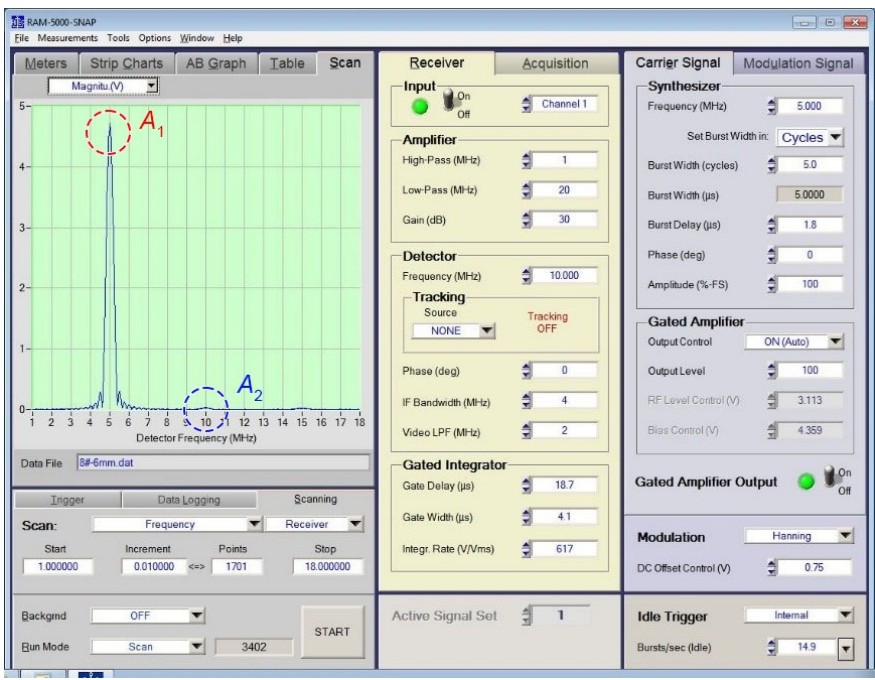

**Figure 4.** Frequency spectrum curve of the received signal of group A4 specimen.

From Figure 4, both the 5 MHz fundamental wave ($A_1$) and the 10 MHz second harmonic wave ($A_2$) were captured by the receiver. The amplitude of fundamental wave ($A_1$) is nearly 4.5 V, whereas the amplitude of the second harmonic wave ($A_2$) is less than 0.1 V. This indicates that the acoustics nonlinearity of plastically deformed 35CrMoA

steel is relatively weak, and generating an observable second harmonic wave requires a high-power fundamental wave.

## 3. Mathematical and Finite Element Models

### 3.1. Theory of Dislocation-Induced Acoustics Nonlinearity

The plastic damage in metal materials is closely related to the evolution of dislocation structure. When subjected to an external load, dislocations within the material nucleate, accumulate and become entangled, resulting in changes to ultrasonic nonlinear parameters. Gliding dislocations are impeded by impurity atoms and precipitation phase particles within the material. Under stress, the dislocation line between two adjacent pinning points deforms into a bow-like string. When a periodic stress wave is applied to this dislocation string, it undergoes forced vibration, leading to waveform distortion of the incident wave and the generation of high-order harmonics.

The propagation of ultrasonic waves generates a small oscillatory stress with an amplitude of $\Delta\sigma$ in addition to the applied stress and results in an additional stran $\Delta\varepsilon$. The relation between the additional stress $\Delta\sigma$ and the additional strain $\Delta\varepsilon$ caused by elastic wave in crystal can be expressed as [30]:

$$\begin{aligned}\Delta\sigma &= \frac{\partial\sigma}{\partial\varepsilon}(\Delta\varepsilon) + \frac{1}{2}\frac{\partial^2\sigma}{\partial\varepsilon^2}(\Delta\varepsilon)^2 + \dots \\ &= A(\Delta\varepsilon) + \frac{1}{2}B(\Delta\varepsilon)^2 + \dots\end{aligned} \tag{8}$$

where

$$A = \left[\frac{1}{A_2^H} + \frac{4(1-v)}{3}\cdot\frac{\Omega\Lambda L^2 R}{\mu}\left(1 + vf_s - 2vf_e\right)^{-1}\right]^{-1}$$

$$B = \frac{-2\left[-\frac{1}{2}\frac{A_3^H}{\left(A_2^H\right)^3} + \frac{12}{5}(1-v)^3\cdot\frac{\Omega AL^4 R^3}{\mu^3 b^2}\cdot\sigma_b\cdot(1+vf_s-2vf_e)^{-3}\right]}{\left[\frac{1}{A_2^H} + \frac{4(1-v)}{3}\cdot\frac{\Omega AL^2 R}{\mu}\left(1+vf_s-2vf_e\right)^{-1}\right]^3} \tag{9}$$

where $\mu$ is the shear modulus, $b$ is the modulus of the Burgers vector, $v$ is Poisson's ratio, $\Lambda$ is the dislocation density, $L$ is half of the length of the dislocation string, $\sigma_b$ is the back stress, $\Omega$ and $R$ are the conversion factors of shear strain and shear stress to longitudinal strain and longitudinal stress respectively. $A_2^H$ and $A_3^H$ are the second-order and third-order Huang coefficients of the lattice respectively, and $f_e$ and $f_s$ are the fractions of edge and screw dislocations in total dislocations, respectively. Ultrasonic nonlinear coefficient $\beta$ can be expressed as:

$$\beta = -\frac{B}{A} = \frac{-\frac{A_3^H}{\left(A_2^H\right)^3} + \frac{24}{5}(1-v)^3\cdot\frac{\Omega AL^4 R^3}{\mu^3 b^2}\cdot\sigma_b\cdot\left(1+vf_s-2vf_e\right)^{-3}}{\left[\frac{1}{A_2^H} + \frac{4(1-v)}{3}\cdot\frac{\Omega AL^2 R}{\mu}\left(1+vf_s-2vf_e\right)^{-1}\right]^2} \tag{10}$$

In most cases, $4(1-v)\Omega\Lambda L^2 R(1+vf_s-2vf_e)^{-1}/3\mu$ is much smaller than $1/A_2^H$ and can be neglected. Therefore, the ultrasonic nonlinear coefficient $\beta$ can be further simplified as

$$\beta = -\frac{A_3^H}{A_2^H} + \frac{192(1-v)^3}{5}\frac{\Lambda L^4\Omega R^3\left(A_2^H\right)^2}{\mu^3 b^2}\times\sigma_b(2-v+3v\cos 2\theta)^{-3} \tag{11}$$

where $\theta$ is the angle between the dislocation line and the Burgers vector.

### 3.2. Dislocation Density Evolution Model

The plastic deformation of metal materials is related to the dislocation motion. According to the Kocks–Mecking model [31], the flow stress $\sigma_s$ is solely related to dislocation density $\Lambda$, which can be expressed as:

$$\sigma_s = \sigma_0 + \alpha\mu b\sqrt{\Lambda}\,(1) \tag{12}$$

where $\alpha$ is a constant, which is 0.5 for most metals. During plastic deformation, the change rate of dislocation density is given as

$$\frac{d\Lambda}{d\varepsilon} = k_1\sqrt{\Lambda} - k_2\Lambda\,(2) \tag{13}$$

where $k_1$ and $k_2$ are dislocation formation and annihilation coefficients, respectively. Dislocation annihilation is the result of dynamic recovery process, and its effect can be neglected in room temperature tensile. The coefficient $k_1$ can be obtained by fitting the flow curve of 35CrMoA steel.

### 3.3. Finite Element Simulation

In the current investigation, the commercial finite element software COMSOL Multiphysics (V6.2, COMSOL Inc., Stockholm, Sweden) was used to model the constitutive equation of acoustics nonlinearity and simulate the propagation of ultrasonic nonlinearity in a plastically deformed 35CrMoA steel specimen. In homogeneous isotropic media, ultrasonic nonlinearity arises from material nonlinearity, geometric nonlinearity, and plastic deformation. It is generally believed that the ultrasonic nonlinearity caused by plastic damage mainly results from the evolution of dislocations in materials. The finite element model is shown in Figure 5.

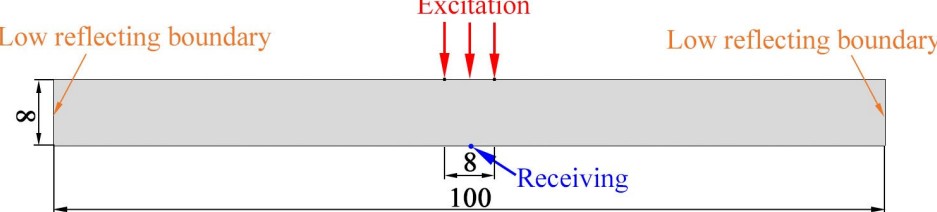

**Figure 5.** Schematic diagram of the simulation model (Unit: mm).

The top and bottom sides, except for the area where the excitation signal is applied, are stress-free and non-reflecting boundary conditions are prescribed for the lateral sides. The fundamental wave was excited by imposing longitudinal stress loadings with a 5-cycle Hanning-windowed sinusoidal tone burst at 5.0 MHz. The width of the excited area is 8 mm (identical to the diameter of the transducers) and the amplitude of the fundamental wave is 20 MPa. The received signal was obtained by extracting the longitudinal displacement at the center of the bottom edge. Subsequently, a Fast Fourier transformation (FFT) was performed on the extracted time domain signal to get the amplitudes of the fundamental and second harmonic signals. Finally, the acoustic nonlinearity parameters can be calculated using Equation (7).

The fourth-order Lagrange element was used in the FE model. To ensure the convergence of calculation, each wavelength should contain at least approximately 1.5 grid elements [32]. The wavelength of an ultrasonic longitudinal wave in steel is approximately 1.2 mm, so the grid size of finite element model elements is selected as 0.8 mm. According to the Fourier or von Neumann stability analysis [33], the time step, $\Delta t$, should be smaller

than the time required for the longitudinal wave to travel across the element length, which is expressed as:

$$\Delta t \leq \frac{\Delta y}{c_L} \tag{14}$$

where $c_L$ is the longitudinal wave velocity in steel and $\Delta y$ is the length of element. The material parameters used in the calculation were $A_2^H = -3A_3^H = 195$ GPa, $\mu = 82$ GPa, $\upsilon = 0.3$, $b = 2.5$ nm, $L = 23$ nm, and the conversion factors $\Omega$ and $R$ were chosen as 0.33 [34]. The edge dislocation and the screw dislocation were assumed to each make up half of the total dislocations. Figure 6 shows the propagation of the ultrasonic wave in the specimen when the plastic strain $\varepsilon = 0.112$.

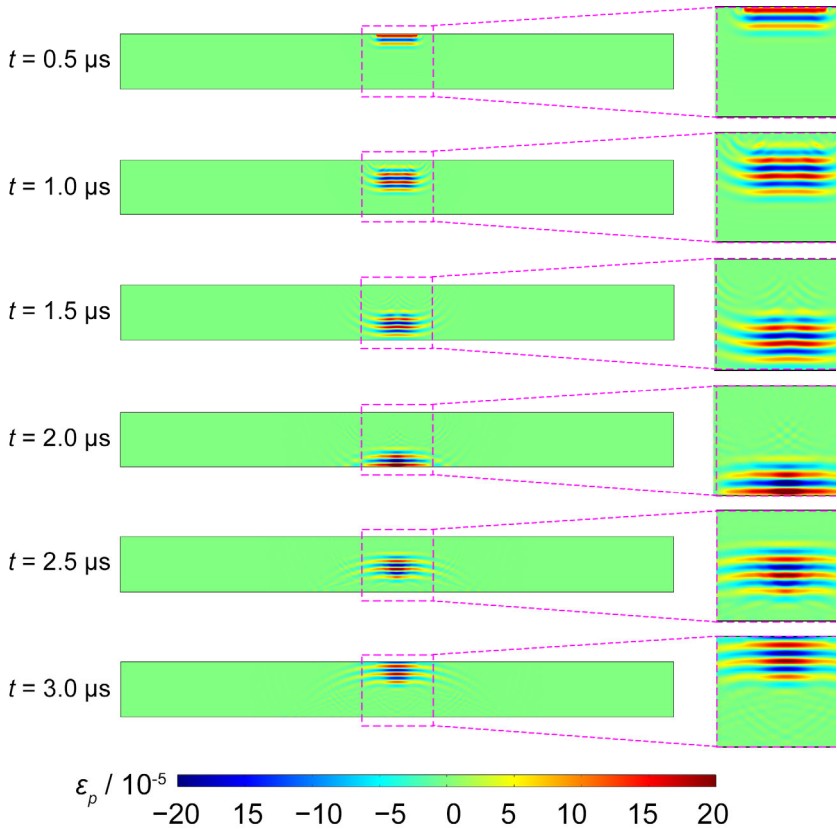

$\varepsilon_p / 10^{-5}$    −20   15   −10   −5   0   5   10   15   20

**Figure 6.** Propagation of elastic waves in the specimen.

It can be seen that the ultrasonic waves propagate to the bottom edge of the specimen after about $t = 1.4$ μs, and reflect several times on the bottom and top edges. For the simulated specimens in this investigation, the ultrasonic nonlinearity mainly arises from the material nonlinearity. The received signal in the time and frequency domain of the received signal in Figure 6 is shown in Figure 7.

The first wave is the signal that the transmitted signal directly reaches the receiving probe on the bottom edge, while the second wave is the signal that the transmitted signal reaches the receiving probe after being reflected once each from the bottom and top edges of the specimen, respectively. The Hanning window is used to intercept the second wave ($t = 4$–$5.5$ μs) of the received signal in Figure 7a, and then the Fast Fourier transform (FFT) was performed on this signal to obtain the spectrum distribution of the received signal. It can be seen that due to the nonlinear stress–strain relationship of the material, the second harmonic at $2f_0$ frequency is generated in the received signal.

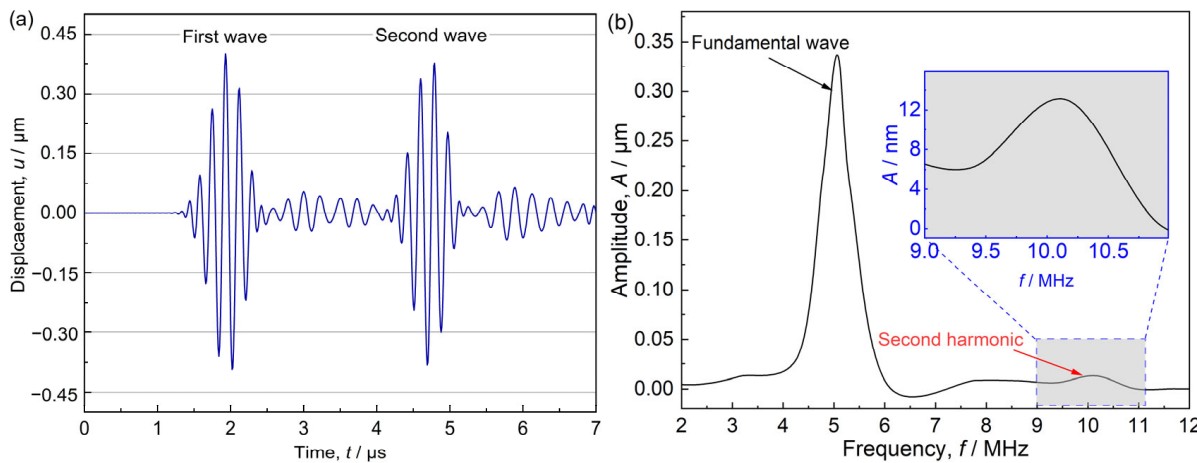

**Figure 7.** Received signal in time (**a**) and frequency (**b**) domain.

## 4. Discussion

The normalized relative nonlinearity parameters of the plastically deformed specimens with respect to the ultrasonic nonlinear parameters $\beta_0$ of the original unstretched specimens were defined as a function of the plastic strain. The results from experimental measurement and finite element simulation are shown in Figure 8.

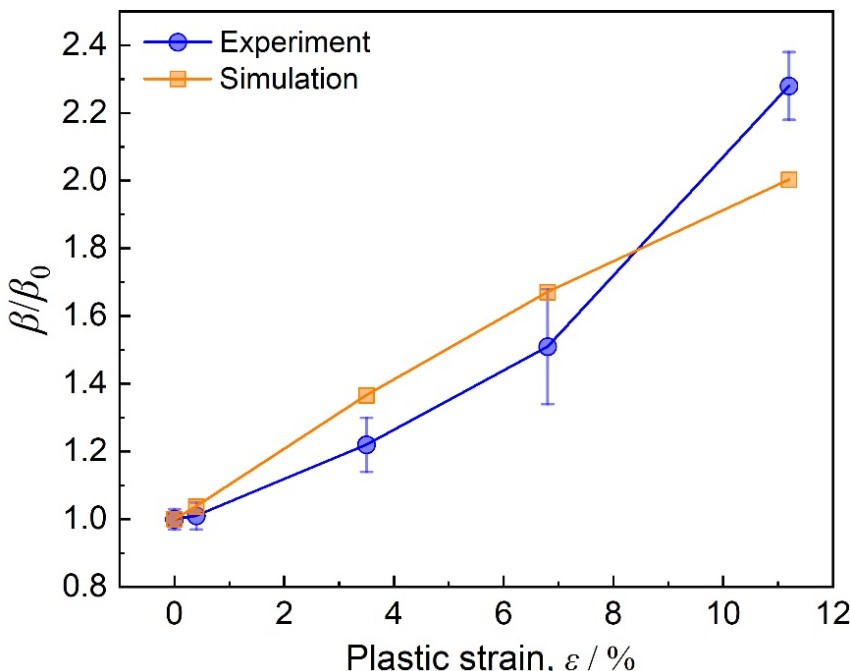

**Figure 8.** Effect of plastic strain on normalized nonlinearity parameter.

From Figure 8, it is evident that the ultrasonic nonlinear coefficient increases with the increase in plastic deformation. In the early stage of plastic deformation, the rise in the ultrasonic nonlinear coefficient can be mainly attributed to the increase in dislocation density. However, in the late stage of plastic deformation, the increase of the ultrasonic nonlinear coefficient is primarily associated with the formation of dislocation walls and dislocation cells [29].

The simulated ultrasonic nonlinear parameters exhibit a two-stage deviation pattern compared to experimental measurements. In the low-strain regime (plastic strain < 8.5%), the finite element (FE) predictions systematically underestimate experimental values. This deviation likely originates from dislocation entanglement mechanisms in high-density

regions, where mutually trapped dislocations restrict their mobility and thereby suppress dislocation-induced ultrasonic nonlinearity [10]. In contrast, a crossover phenomenon emerges at plastic strains exceeding 8.5%, with the FE model overestimating the experimental ultrasonic nonlinear parameters. This reversal correlates with microstructural evolution during plastic deformation: Progressive dislocation accumulation leads to self-organization into cellular structures (dislocation walls/cells) that demonstrate enhanced acoustic nonlinearity compared to planar dislocation configurations [29]. Notably, the current modeling framework exclusively accounts for planar dislocation dynamics, failing to capture the diminished nonlinear response associated with these three-dimensional dislocation structures. This modeling limitation explains the increasing discrepancy at advanced deformation stages.

## 5. Conclusions

Investigations into the propagation of longitudinal waves and the ultrasonic nonlinear response of plastically deformed 35CrMoA steel were conducted through experiments and finite element simulations, ultilizing the dislocation-induced acoustic nonlinearity theory as a basis. The following conclusions have been drawn:

(1) Simulated and experimental results show a strong correlation between the acoustic nonlinearity parameter and plastic strain. Both simulated and experimental results demonstrate a monotonic increase in the nonlinear acoustic parameter with applied plastic strain.

(2) The simulation results for the ultrasonic nonlinear coefficient are lower than the experimental results when the plastic strain is lower than about 8.5%. This difference may be attributed to the dislocation entanglement in the areas with high dislocation density during the early stage of plastic deformation, which hinders dislocation movement and consequently reduces the nonlinear ultrasonic response of dislocations.

(3) When the plastic strains exceed about 8.5%, as dislocation density increases, the dislocations pile up with each other, forming dislocation cells and dislocation walls. The ultrasonic nonlinear response induced by dislocation cells and dislocation walls is higher than that of plane dislocations. Therefore, the calculated results for ultrasonic nonlinear coefficients in the later stage of plastic deformation are lower than the experimental results.

**Author Contributions:** Conceptualization, X.Y.; Methodology, S.Y.; Software, X.J.; Validation, L.H.; Investigation, S.Y. and X.J.; Resources, L.H.; Writing—original draft, S.Y.; Writing—review & editing, L.H.; Supervision, X.Y. All authors have read and agreed to the published version of the manuscript.

**Funding:** This research received no external funding.

**Data Availability Statement:** The original contributions presented in this study are included in the article. Further inquiries can be directed to the corresponding author.

**Conflicts of Interest:** Author Shumin Yu, Xinbin Yang are employed by the company Zhengzhou Guodian Mechanical Design Institute Co., Ltd. The remaining authors declare that the research was conducted in the absence of any commercial or financial relationships that could be construed as a potential conflict of interest.

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
