# Peer review of "Finite Element Modeling of Acoustic Nonlinearity Derived from Plastic Deformation of 35CrMoA Steel"

_metals, doi:10.3390/met15040343_

Round 1
Reviewer 1 Report
Comments and Suggestions for Authors
Dear Authors,
I find Your paper very interesting. It is well structured, and has a valuable contribution to the nonlinear ultrasonics and material science.
Please check for some minor grammatical errors and typing errors.
Also, please check Equation 4. for any errors.
Regarding Figure 8., the experimental results are compared to numerical simulation, it is evident that these two curves do not coincide, that is, there are some deviations. This is explained in following paragraph, but also mentioned: "During the early stage of plastic deformation, the simulation results are lower than the experimental results." This is not quite clear from the Figure 8. Can You explain this a little bit more thoroughly?
Other then this, I don't have any more comments.
Regards!
Author Response
Thank you very much for taking the time to review this manuscript. We sincerely appreciate your constructive feedback. Please find our detailed point-by-point responses below. All corresponding revisions have been highlighted in the resubmitted manuscript for your convenience.
Comments 1: Please check for some minor grammatical errors and typing errors.
Response: Thanks again for your careful reading of the manuscript. We have tried our best to avoid any grammatical mistakes or improper descriptions in our revised manuscript, which highlights all the revised parts. We hope the language is now easy to understand for potential readers.
Comments 2: Also, please check Equation 4. for any errors.
Response: Equation 4 has been revised as follows [1, 2]:
[1] A Hikata, B B Chick, C Elbaum. Dislocation contribution to the second harmonic generation of ultrasonic waves. Journal of Applied Physics, 1965, 36(1): 229-236.
[2] K Y Jhang, K J Kim. Evaluation of material degradation using nonlinear acoustic effect. Ultrasonics, 1999, 37(1): 39-44
Comments 3:Regarding Figure 8., the experimental results are compared to numerical simulation, it is evident that these two curves do not coincide, that is, there are some deviations. This is explained in following paragraph, but also mentioned: "During the early stage of plastic deformation, the simulation results are lower than the experimental results." This is not quite clear from the Figure 8. Can You explain this a little bit more thoroughly?
Response: We have revised the paragraph following Figure 8 to give a more thoroughly explanation. The paragraph has been revised as follows:
“The simulated ultrasonic nonlinear parameters exhibit a two-stage deviation pattern compared to experimental measurements. In the low-strain regime (plastic strain <8.5%), the finite element (FE) predictions systematically underestimate experimental values. This deviation likely originates from dislocation entanglement mechanisms in high-density regions, where mutually trapped dislocations restrict their mobility and thereby suppress dislocation-induced ultrasonic nonlinearity [10]. In contrast, a crossover phenomenon emerges at plastic strains exceeding 8.5%, with the FE model overestimating the experimental ultrasonic nonlinear parameters. This reversal correlates with microstructural evolution during plastic deformation: Progressive dislocation accumulation leads to self-organization into cellular structures (dislocation walls/cells) that demonstrate enhanced acoustic nonlinearity compared to planar dislocation configurations [29]. Notably, the current modeling framework exclusively accounts for planar dislocation dynamics, failing to capture the diminished nonlinear response associated with these three-dimensional dislocation structures. This modeling limitation explains the increasing discrepancy at advanced deformation stages.”
Reviewer 2 Report
Comments and Suggestions for Authors
The main question addressed by the research is the use of finite element model based on the dislocation-induced ultrasonic nonlinearity theory to simulate the ultrasonic nonlinearity of plastically deformed material.
The subject is original for this field, being an extremely efficient method to study the plastic deformation of metals.
This method uses the simulation of the ultrasonic nonlinear coefficients of material at different plastic damages, in order to obtain the relationship between plastic damage, dislocation evolution and ultrasonic nonlinearity of the material.
The experimental validation was carried out to verify the accuracy of the proposed model. After the presentation of the samples and the experimental setup, the mathematical model was developed.
The research method is adequate. The simulated results provided the theoretical foundation for the ultrasonic nonlinear evaluation of plastic damage of the metal material.
The conclusions are consistent, both simulated and experimental results demonstrated a strong correlation between the acoustic nonlinearity parameter and plastic strain, but the simulated nonlinearity parameters were underestimated in the later stages of the plastic deformation.
The tables and figures are clear, with explicit legends. The authors used 29 appropriate references.
Observations:
Lines 193-194 “The width of the excited aera is 8 mm (identical to the diameter of transducers)” is in contradiction with the lines 91-92 “To generate and measure the longitudinal wave, two commercial piezoelectric transducers with a diameter of Φ = 6 mm were positioned at the center of the top and bottom surfaces of the specimens.”
Line 225: The Hhanning window > The Hanning window
Author Response
Thank you very much for taking the time to review this manuscript. We sincerely appreciate your constructive feedback. Please find our detailed point-by-point responses below. All corresponding revisions have been highlighted in the resubmitted manuscript for your convenience.
Comments 1:Lines 193-194 “The width of the excited aera is 8 mm (identical to the diameter of transducers)” is in contradiction with the lines 91-92 “To generate and measure the longitudinal wave, two commercial piezoelectric transducers with a diameter of Φ = 6 mm were positioned at the center of the top and bottom surfaces of the specimens.”
Response: We're sorry we made a typo here. The diameter of each transducer is Φ = 8 mm. The manuscript has been revised accordingly.
Comments 2: Line 225: The Hhanning window > The Hanning window
Response: The typo has been corrected.
Reviewer 3 Report
Comments and Suggestions for Authors
The authors need to address the following queries.
- There are several grammatical and typographical errors throughout the manuscript. It requires thorough proofreading to enhance readability. For example, Line 7: "Acoustic nonlinearity is derivred..." → "Acoustic nonlinearity is derived..."
- The abstract needs to be revised to improve the summary of findings and emphasize key contributions.
- Line 42-46: Provide more details on why previous models were insufficient. Clearly state the novelty of this study compared to prior work.
- A few equations (e.g., Eq. 7 and Eq. 11) would need to be better described with definitions for each term.
- Although the manuscript cites related works, more extensive discussion on past research applying FEM to nonlinear ultrasonic testing would be helpful.
- Try citing more recent publications on ultrasonic nonlinearity in other materials.
- More information regarding the calibration procedure of the ultrasonic testing system would be beneficial.
- How were the measurement uncertainties addressed? A few words on error estimation would make the results more robust.
- The authors hypothesize the predominance of dislocation effects in the later stages of plastic deformation. Experimental evidence indicates a larger contribution of dislocation cell and wall structures. Can the FEM model be adapted to consider these structures?
- References or experimental results must support the selection of material parameters (e.g., dislocation density values).
- Comparing the results obtained with other studies on various grades of steel or alloys would be helpful to prove the results' extended applicability.
- Ensure all figure captions provide sufficient context without referring to the main text.
- Kindly reconcile the conclusion with the study objectives.
- What are the practical implications of this study and the future directions? Kindly state?
There are several grammatical and typographical errors throughout the manuscript. It requires thorough proofreading to enhance readability.
Round 2
Reviewer 3 Report
Comments and Suggestions for Authors
The authors have addressed all the queries. The article may be accepted in its present form.